# Statistical Mechanics of Min-Max Problems

**Yuma Ichikawa**                    *ichikawa-yuma1@g.ecc.u-tokyo.ac.jp, ichikawa.yuma@fujitsu.com*
*Department of Basic Science, University of Tokyo*
*Fujitsu Limited*

**Koji Hukushima**                    *k-hukushima@g.ecc.u-tokyo.ac.jp*
*Department of Basic Science, University of Tokyo*

**Reviewed on OpenReview:** *https://openreview.net/forum?id=qZqUFeTtuI*

## Abstract

Min-max optimization problems, also known as saddle point problems, have attracted significant attention due to their applications in various fields, such as fair beamforming, generative adversarial networks (GANs), and adversarial learning. However, understanding the properties of these min-max problems has remained a substantial challenge. This study introduces a statistical mechanical formalism for analyzing the equilibrium values of min-max problems in the high-dimensional limit, while appropriately addressing the order of operations for min and max. As a first step, we apply this formalism to bilinear min-max games and simple GANs, deriving the relationship between the amount of training data and generalization error and indicating the optimal ratio of fake to real data for effective learning. This formalism provides a groundwork for a deeper theoretical analysis of the equilibrium properties in various machine learning methods based on min-max problems and encourages the development of new algorithms and architectures.

## 1 Introduction

Min-max optimization problems, also known as saddle point problems, are well-known classical optimization problems extensively studied in the context of zero-sum games (Wald, 1945; Von Neumann & Morgenstern, 1947). These problems have diverse applications across various fields, such as game theory, machine learning, and signal processing. In game theory, min-max problems arise in zero-sum games where one player's gain corresponds to another's loss. Several methods have been proposed to find the min-max value or equilibrium points in these games (Dem'yanov & Pevnyi, 1972; Maistroskii, 1977; Bruck, 1977; Lions, 1978; Nemhauser & Wolsey, 1988; Freund & Schapire, 1999). In machine learning, min-max games are relevant for training generative adversarial networks (GANs) (Goodfellow et al., 2020; Arjovsky et al., 2017), Additionally, in adversarial learning, these problems are employed to train models that are robust to adversarial attacks by optimizing a worst-case perturbed loss function (Szegedy et al., 2013; Goodfellow et al., 2014b; Papernot et al., 2016; Madry et al., 2017),

Despite the widespread application of min-max optimization problems, several challenges still need to be addressed, including understanding the usefulness of these min-max formulations, evaluating the convergence properties of the algorithms, and conducting sensitivity analyses of min-max values. A promising approach to addressing these issues is to analyze the typical-case behavior of min-max problems by examining the min-max value averaged over random instances drawn from distributions that capture realistic settings, referred to as *randomized instance ensembles*. Statistical-mechanical approaches, which have demonstrated their effectiveness in analyzing the typical-case behavior of randomized instance ensembles of optimization and constraint-satisfaction problems (Mézard & Parisi, 1986; Fontanari, 1995), provide a powerful formalism for such analyses. Extending this formalism to analyze the typical-case behavior of min-max values thus presents a potential direction for further research, although this has not yet been fully explored.

This study applies the statistical mechanical formalism to min-max problems, modeling them as a virtual two-temperature system. This formalism enables a sensitivity analysis of the typical-case min-max values in the high-dimensional limit. Notably, this formalism properly addresses the order of min-max operations, critical in non-convex scenarios where interchanging the order of min and max can lead to incorrect results (Razaviyayn et al., 2020). Using this formalism, we analyze typical-case min-max values of bilinear min-max games and simple GANs. In particular, we derive the relationship between the amount of training data and generalization error and indicate the optimal ratio of fake data to real data for effective learning.

Our main contributions are as follows:

- We introduce a statistical-mechanical formalism developed for sensitivity analysis of equilibrium values in high-dimensional min-max problems.

- Applying this approach, we conduct a detailed sensitivity analysis on a bilinear min-max game to verify the theoretical validity of our approach.

- Building on this formalism, we analyze the generalization performance of GANs and determine the optimal ratio between fake and real data for practical training.

## 2 Related Work

The replica method, which is employed in this study, is a non-rigorous but powerful heuristic approach in statistical physics (Edwards & Anderson, 1975; Mézard et al., 1987; Mezard & Montanari, 2009). It has been proven to be a valuable method for high-dimensional machine-learning problems. Previous studies have investigated the relationship between dataset size and generalization error in supervised learning, including single-layer (Gardner & Derrida, 1988; Opper & Haussler, 1991; Barbier et al., 2019; Aubin et al., 2020) and multi-layer (Aubin et al., 2018) neural networks, as well as kernel methods(Dietrich et al., 1999; Bordelon et al., 2020; Gerace et al., 2020). In unsupervised learning, the replica method has also been applied to dimensionality reduction techniques such as the principal component analysis (Biehl & Mietzner, 1993; Hoyle & Rattray, 2004; 2007), and to generative models such as energy-based models (Decelle et al., 2018; Ichikawa & Hukushima, 2022) and denoising autoencoders (Cui & Zdeborová, 2023). However, the dataset-size dependence of GANs has not been previously analyzed, which this study aims to address.

Related to our work, a statistical mechanical formalism for addressing min-max problems has been proposed (Varga, 1998). However, the treatment of the inverse temperature limit differs from our approach, and it has limitations in accurately handling the order of the min and max operations. In the context of adversarial learning, which involves a non-convex and concave min-max problem, Tanner et al. (2024) analyzes a tractable setting where the internal maximization can be solved. By reducing such cases to standard optimization problems, they apply the replica method and approximate message passing to explore the core phenomenology observed in the adversarial robustness. Even in cases where the internal maximization cannot be explicitly solved, the formalism discussed here provides a basis for further analysis and potential extensions to more complex scenarios.

**Notation** Here, we summarize the notations used in this study. We use the shorthand expression $[N] = \{1, 2, \ldots, N\}$, where $N \in \mathbb{N}$. $I_d \in \mathbb{R}^{d \times d}$ denotes a $d \times d$ identity matrix, and $\mathbf{1}_d$ denotes the vector $(1, \ldots, 1)^\top \in \mathbb{R}^d$ and $\mathbf{0}_d$ denotes the vector $(0, \ldots, 0)^\top \in \mathbb{R}^d$. For a matrix $A = (A_{ij}) \in \mathbb{R}^{d \times k}$ and a vector $\boldsymbol{a} = (a_i) \in \mathbb{R}^d$, we use the shorthand expressions $dA \triangleq \prod_{i=1}^{d} \prod_{j=1}^{k} dA_{ij}$ and $d\boldsymbol{a} \triangleq \prod_{i=1}^{d} da_i$, respectively. The notation $\mathcal{O}_{d_x, d_y}(1)$ describes the asymptotic order of a function with respect to the parameters $d_x$ and $d_y$. Specifically, a function $f(d_x, d_y)$ is said to be $\mathcal{O}_{d_x, d_y}(1)$ if it remains bounded as $d_x$ and $d_y$ grow large (or tend toward some specified limit), independently of $d_x$ and $d_y$. The standard Gaussian measure is defined as $D\boldsymbol{z} \triangleq d\boldsymbol{z} e^{-\|\boldsymbol{z}\|^2/2}/(2\pi)^{n/2}$. The notation $\text{extr}_{\boldsymbol{x}} f(\boldsymbol{x})$ represents the evaluation of a function $f(\boldsymbol{x})$ at its extremum with respect to the variable $\boldsymbol{x}$. Specifically, this shorthand implies locating and evaluating $f(\boldsymbol{x})$ at points where its gradient $\nabla_{\boldsymbol{x}} f(\boldsymbol{x}) = \mathbf{0}$.

# 3 Statistical Physics Formalism for Min-Max Optimization Problems

This section introduces a statistical-mechanical formalism that models min-max problems as a virtual two-temperature system from a statistical mechanics perspective. Min-max problems are formally expressed as

$$\Psi(A) = \min_{\boldsymbol{x} \in \mathcal{X}} \max_{\boldsymbol{y} \in \mathcal{Y}} V(\boldsymbol{x}, \boldsymbol{y}; A), \quad \text{s.t.} \quad \boldsymbol{x} \in \mathcal{X} \subseteq \mathbb{R}^{d_x}, \quad \boldsymbol{y} \in \mathcal{Y} \subseteq \mathbb{R}^{d_y}, \tag{1}$$

where $V(\cdot, \cdot) : \mathbb{R}^{d_x} \times \mathbb{R}^{d_y} \to \mathbb{R}$ is a bivariate function; $\boldsymbol{x} \in \mathbb{R}^{d_x}$ and $\boldsymbol{y} \in \mathbb{R}^{d_y}$ are the optimization variables; $\mathcal{X}$ and $\mathcal{Y}$ are the feasible sets; $A$ is a parameter characterizing the problem, e.g., graph $G$. We introduce the following Boltzmann distribution to analyze min-max problems for a given bivariate function $V(\boldsymbol{x}, \boldsymbol{y}; A)$ in Eq. (1), with virtual inverse temperatures $\beta_{\min} \in \mathbb{R}$ and $\beta_{\max} \in \mathbb{R}$:

$$p_{\beta_{\min}, \beta_{\max}}(\boldsymbol{x}; A) \triangleq \frac{1}{\mathcal{Z}(\beta_{\min}, \beta_{\max}, A)} e^{-\beta_{\min} \left( \frac{1}{\beta_{\max}} \ln \int_{\mathcal{Y}} d\boldsymbol{y} e^{\beta_{\max} V(\boldsymbol{x}, \boldsymbol{y}; A)} \right)},$$

where $\mathcal{Z}(\beta_{\min}, \beta_{\max}, A)$ is the normalization constant, also known as the partition function. Hereafter, we refer to it as the partition function. In this context, a two-temperature system is particularly important because it allows us to distinguish between the opposing optimization objectives inherent in min-max problems, similar to the different thermal behaviors in statistical mechanics.

By taking the limit $\beta_{\max} \to +\infty$ followed by $\beta_{\min} \to +\infty$, the distribution $\lim_{\beta_{\min} \to +\infty} \lim_{\beta_{\max} \to +\infty} p_{\beta_{\min}, \beta_{\max}}(\boldsymbol{x}; A)$ concentrates on a uniform distribution over the min-max values, assuming that well-defined min-max values exist for $V(\cdot, \cdot)$ and the min-max value is bounded over feasible sets $\mathcal{X}$ and $\mathcal{Y}$. Note that the order of these limits is crucial because the min and max operations cannot be interchanged in non-convex and non-concave min-max problems (Razaviyayn et al., 2020), i.e., $\min_{\boldsymbol{x} \in \mathcal{X}} \max_{\boldsymbol{y} \in \mathcal{Y}} V(\boldsymbol{x}, \boldsymbol{y}; A) \neq \max_{\boldsymbol{y} \in \mathcal{Y}} \min_{\boldsymbol{x} \in \mathcal{X}} V(\boldsymbol{x}, \boldsymbol{y}; A)$. While a similar formulation has been used in the previous work (Varga, 1998), they simultaneously take the limits of both $\beta_{\min}$ and $\beta_{\max}$ with a fixed ratio $\beta_{\min}/\beta_{\max} = \mathcal{O}_{\beta_{\min}, \beta_{\max}}(1)$, which does not fully capture the distinct effects of min and max operations in non-convex settings. Such an approach generally does not yield accurate results when the function $V(\boldsymbol{x}, \boldsymbol{y}; A)$ is non-convex with respect to $\boldsymbol{x}$ and $\boldsymbol{y}$.

Statistical-mechanical approaches have demonstrated their effectiveness in analyzing the typical-case behavior specifically, the properties of the optimal value averaged over the instances that follow a distribution $p(A)$– for optimization and constraint-satisfaction problems (Mézard & Parisi, 1986; Fontanari, 1995). These analyses have succeeded in providing insights into different aspects of combinatorial optimization, unlike worst-case analysis. This work also focuses on evaluating the typical cases of min-max problems characterized by a random parameter $A$. Our main objective is to calculate the logarithm of $Z(\beta_{\min}, \beta_{\max}, A)$ averaged over the random variables $A$ in the limit $\beta_{\max} \to \infty$ followed by $\beta_{\min} \to \infty$:

$$\Omega = \lim_{\beta_{\min} \to \infty} \lim_{\beta_{\max} \to \infty} f(\beta_{\min}, \beta_{\max}),$$

where

$$f(\beta_{\min}, \beta_{\max}) \triangleq -\frac{1}{\beta_{\min} d_x} \mathbb{E}_A \left[ \log \mathcal{Z}(\beta_{\min}, \beta_{\max}, A) \right],$$

which is referred to as the free energy density. This free energy is a generating function with variables $\boldsymbol{x}$ and $\boldsymbol{y}$. Appendix C shows how to calculate the function of the optimal value $\boldsymbol{x}$ through this free energy.

Setting the ratio of the inverse temperatures as $p = -\beta_{\min}/\beta_{\max}$, this can be rewritten as

$$f(\beta_{\min}, \beta_{\max}) = -\frac{1}{\beta_{\min} d_x} \mathbb{E}_A \left[ \log \int_{\mathcal{X}} d\boldsymbol{x} e^{-\beta_{\min} \left( \frac{1}{\beta_{\max}} \ln \int d\boldsymbol{y} e^{\beta_{\max} V(\boldsymbol{x}, \boldsymbol{y}; A)} \right)} \right],$$

$$= -\frac{1}{\beta_{\min} d_x} \mathbb{E}_A \left[ \log \int_{\mathcal{X}} d\boldsymbol{x} \left( \int_{\mathcal{Y}} d\boldsymbol{y} e^{\beta_{\max} V(\boldsymbol{x}, \boldsymbol{y}; A)} \right)^p \right]. \tag{2}$$

Although calculating the expectation value of the logarithm is generally difficult, we begin by using the identity

$$\mathbb{E}_A[\log f(A)] = \lim_{\gamma \to +0} \frac{1}{\gamma} \log \mathbb{E}_A[(f(A))^\gamma] \tag{3}$$

to expand the logarithmic form as follows:

$$\Omega = -\lim_{\beta_{\min}\to\infty}\lim_{\beta_{\max}\to\infty}\lim_{\gamma\to+0}\frac{1}{\beta_{\min}d_x\gamma}\log\tilde{Z}_\gamma(\beta_{\min},\beta_{\max}),\tag{4}$$

where

$$\tilde{Z}_\gamma(\beta_{\min},\beta_{\max}) \triangleq \mathbb{E}_A\left[\left(\left(\int_\mathcal{X}d\boldsymbol{x}\left(\int_\mathcal{Y}d\boldsymbol{y}e^{\beta_{\max}V(\boldsymbol{x},\boldsymbol{y};A)}\right)^p\right)^\gamma\right]\right].\tag{5}$$

At this point, note that these transformations are purely algebraic identities without assuming the parameters $\gamma$ or $p$ to be integers.

Following the idea of the replica method (Edwards & Anderson, 1975; Parisi, 1979; 1983; Zdeborová & Krzakala, 2016; Gabrié, 2020), we then proceed under the assumption that $\gamma$ and $p$ are natural numbers. Specifically, rather than addressing Eq (4) directly real values $\gamma$ and $p$, one calculates the average of the $\gamma$-th and $p$-th powers for $\gamma, p \in \mathbb{N}$, then performs an analytic continuation to $\gamma, p \in \mathbb{R}$ for this expression, and finally takes the limits $\gamma \to +0$, $\beta_{\min} \to +\infty$ and $\beta_{\max} \to +\infty$. Based on this replica "trick", the calculation simplifies to the replicated partition function $Z_\gamma(\beta_{\min},\beta_{\max})$ as an approximation of $\tilde{Z}_\gamma(\beta_{\min},\beta_{\max})$:

$$\tilde{Z}_\gamma(\beta_{\min},\beta_{\max}) \approx Z_\gamma(\beta_{\min},\beta_{\max}) \triangleq \mathbb{E}_A\left[\prod_{a=1}^\gamma\int_{\mathcal{X}^a}d\boldsymbol{x}^a\prod_{l=1}^p\int_{\mathcal{Y}^{al}}d\boldsymbol{y}^{al}e^{\beta_{\max}\sum_{a,l}V(\boldsymbol{x}^a,\boldsymbol{y}^{al};A)}\right],\tag{6}$$

up to the first order of $\gamma$ to take the $\gamma \to +0$ limit on the right-hand side of Eq. (4). This computation is a standard procedure in the statistical physics of interaction systems including random variables, and is generally accepted as exact, although rigorous proof has not yet been provided. Specifically, the mathematical rigor of the method remains limited due to the unproven uniqueness of the analytic continuation, an issue noted for the moment problem (Tanaka, 2007). As noted in Section 2, the replica method has provided various results in high dimensional statistics and machine learning as well.

Additionally, before taking the limits, $\beta_{\min} \to \infty$ and $\beta_{\max} \to \infty$, the concept of finite inverse temperatures $\beta_{\min}$ and $\beta_{\max}$ corresponds to scenarios where neither the minimum nor the maximum is fully achieved, a common situation in the min-max algorithms. This approach provides valuable insights into cases where neither extreme is fully realized or both are only partially optimized. Exploring novel algorithms based on this finite-temperature generalization of min-max problems represents an intriguing direction for future work. Furthermore, in game theory, this formalism can be interpreted as a framework for modeling games under relaxed assumptions of complete rationality, where players $\boldsymbol{x}$ and $\boldsymbol{y}$ are assumed to behave with bounded rationality rather than adhering strictly to classical models of fully rational behavior (Von Neumann & Morgenstern, 1947).

In the following sections, we apply this formalism to a fundamental and significant bilinear min-max game, demonstrating that the analytic continuation of $p$ in the replica method is a rigorous operation. We then analyze the minimal model of GANs as a more practical example.

## 4 Bilinear Min-max Games

This min-max formalism introduces two replica parameters: $\gamma$, associated with the randomness of $A$, and $p$, related to the dual structure of min-max problems. The analytic continuation with respect to the replica parameter $\gamma$ is widely recognized as effective and is frequently employed in the statistical mechanics of optimization. However, the analytic continuation of the replica parameter $p$ has not yet been explored. While establishing its mathematical validity presents challenges, this study eliminates the influence of the replica parameter $\gamma$ associated with the randomness of $A$ and rigorously demonstrates that the analytic continuation with respect to $p$ holds for fundamental bilinear min-max games. Specifically, we show that the free energy density derived using the replica trick in Eq. (6), as explained in Section 3, is equivalent to the exact expression in Eq. (5) derived without analytic continuation of $\gamma$ and $p$ for bilinear min-max games (Tseng, 1995; Daskalakis et al., 2017).

Bilinear games are regarded as a fundamental example for studying new min-max optimization algorithms and techniques (Daskalakis et al., 2017; Gidel et al., 2019; 2018; Liang & Stokes, 2019). Mathematically,

bilinear zero-sum games can be formulated as the following min-max problem:

$$\min_{\boldsymbol{x} \in \{0,1\}^{d_x}} \max_{\boldsymbol{y} \in \{0,1\}^{d_y}} V(\boldsymbol{x}, \boldsymbol{y}; \boldsymbol{W}),$$

where $V(\cdot, \cdot)$ is given by

$$V(\boldsymbol{x}, \boldsymbol{y}; \boldsymbol{W}) = \frac{1}{2d_x} \boldsymbol{x}^\top W_{xx} \boldsymbol{x} + \frac{1}{2d_y} \boldsymbol{y}^\top W_{yy} \boldsymbol{y} + \frac{1}{\sqrt{d_x d_y}} \boldsymbol{x}^\top W_{xy} \boldsymbol{y} + \boldsymbol{x}^\top \boldsymbol{b}_x + \boldsymbol{y}^\top \boldsymbol{b}_y,$$

where $\boldsymbol{W} = (W_{xx}, W_{yy}, W_{xy}, \boldsymbol{b}_x, \boldsymbol{b}_y)$. For simplicity, we assume $W_{xx} = w_{xx} \mathbf{1}_{d_x \times d_x} \in \mathbb{R}^{d_x \times d_x}$, $W_{xy} = w_{xx} \mathbf{1}_{d_x \times d_y} \in \mathbb{R}^{d_x \times d_y}$, $W_{yy} = w_{yy} \mathbf{1}_{d_y \times d_y} \in \mathbb{R}^{d_y \times d_y}$, $\boldsymbol{b}_x = b_x \mathbf{1}_{d_x} \in \mathbb{R}^{d_x}$, and $\boldsymbol{b}_y = b_y \mathbf{1}_{d_y} \in \mathbb{R}^{d_y}$. The following results can be readily extended to the matrices $W_{xx}$, $W_{yy}$, and $W_{xy}$ with a limited number of eigenvalues of $\mathcal{O}_{d_x, d_y}(1)$. For a detailed discussion, refer to Appendix B.

In this setting, the analytically continued free energy density $f(\beta_{\min}, \beta_{\max}; \boldsymbol{W})$ calculated using replicated partition function $Z_\gamma(\beta_{\min}, \beta_{\max})$ in Eq. (6) coincides with the exact free energy density $\tilde{f}(\beta_{\min}, \beta_{\max}; \boldsymbol{W})$ from the partition function $\tilde{Z}_\gamma(\beta_{\min}, \beta_{\max})$ in Eq. (5).

**Theorem 4.1** *For any $\beta_{\min}, \beta_{\max} \in \mathbb{R}$ and $w_{xx}, w_{xy}, w_{yy}, b_x, b_y \in \mathbb{R}$, the following equality holds:*

$$f(\beta_{\min}, \beta_{\max}; \boldsymbol{W}) = \tilde{f}(\beta_{\min}, \beta_{\max}; \boldsymbol{W}),$$

*where*

$$f(\beta_{\min}, \beta_{\max}; \boldsymbol{W}) = \operatorname*{extr}_{m^x, m^y} \left[ \frac{w_{xx}}{2}(m^x)^2 + \frac{\kappa w_{yy}}{2}(m^y)^2 + w_{xy} \kappa^{1/2} m^x m^y \right.$$

$$\left. + b_x m^x + \kappa b_y m^y - \frac{1}{\beta_{\min}} H(m^x) + \frac{\kappa}{\beta_{\max}} H(m^y) \right],$$

*where $\kappa = {}^{d_y}/_{d_x}$, $H(x) = -x \log(x) - (1-x) \log(1-x)$ denotes binary cross entropy, and* extr *denotes the extremum operation.*

This theorem establishes the validity of the analytic continuation for the replica parameter $p$ using Eq. (6) for bilinear min-max games. The detailed proof of this theorem is provided in Appendix A.

## 5 Generative Adversarial Networks

Generative adversarial networks (GANs) (Goodfellow et al., 2020) aim to model high-dimensional probability distributions based on training datasets. Despite significant progress in practical applications (Arjovsky et al., 2017; Lucic et al., 2018; Ledig et al., 2017; Isola et al., 2017; Reed et al., 2016), several issues are yet to be resolved, including how the amount of training data influences generalization performance and how sensitive GANs are to specific hyperparameters. This section analyzes the relationship between the amount of training data and generalization error. Additionally, we conduct a sensitivity analysis on the ratio of fake data generated by the generator to the amount of training data, which is critical for the training of GANs. Our analysis employs a minimal setup that captures the intrinsic structure and learning dynamics of GANs (Wang et al., 2019). We consider the high-dimensional limit, where the number of real and fake samples, $n$ and $\tilde{n}$, respectively, and the dimension $d$ are large while remaining comparable. Specifically, we analyze the regime in which $n, \tilde{n}, d \to \infty$ while maintaining a comparable ratio, i.e., $\alpha = {}^n/_d = \Theta(d^0)$ and $\tilde{\alpha} = {}^{\tilde{n}}/_d = \Theta(d^0)$, commonly referred to as sample complexity.

### 5.1 Settings

**Generative model for the dataset** We consider that a training dataset $\mathcal{D} = \{\boldsymbol{x}^\mu\}_{\mu=1}^n$, where each $\boldsymbol{x}^\mu \in \mathbb{R}^d$ is drawn from the following distribution:

$$\boldsymbol{x}^\mu = \frac{1}{\sqrt{d}} \boldsymbol{w}^* c^\mu + \sqrt{\eta} \boldsymbol{n}^\mu, \tag{7}$$

where $\boldsymbol{w}^* \in \mathbb{R}^d$ is a deterministic feature vector, $c^\mu \in \mathbb{R}$ is random scalar drawn from a standard normal distribution $p(c) = \mathcal{N}(c; 0, 1)$, $\boldsymbol{n}^\mu$ is a background noise vector whose components are i.i.d. from the standard normal distribution $\mathcal{N}(\boldsymbol{n}; \boldsymbol{0}_d, I_d)$, and $\eta \in \mathbb{R}$ is a scalar parameter to control the strength of the noise. We also assume that $\|\boldsymbol{w}^*\|^2 = 1$. This generative model, known as the spiked covariance model (Wishart, 1928; Potters & Bouchaud, 2020), has been studied in statistics to analyze the performance of unsupervised learning methods such as PCA (Ipsen & Hansen, 2019; Biehl & Mietzner, 1993; Hoyle & Rattray, 2004), sparse PCA (Lesieur et al., 2015), deterministic autoencoders (Refinetti & Goldt, 2022), and variational autoencoder (Ichikawa & Hukushima, 2024; 2023).

**GAN model**   Following Wang et al. (2019), we assume that the generator has the same linear structure as the dataset generative model described in Eq. (7):

$$g(z; \boldsymbol{w}) \triangleq \frac{1}{\sqrt{d}} \boldsymbol{w} z + \sqrt{\tilde{\eta}} \tilde{\boldsymbol{n}}, \tag{8}$$

where $\boldsymbol{w} \in \mathbb{R}^d$ is a learnable parameter, $z \in \mathbb{R}$ is a latent variable drawn from a standard normal distribution $p(z) = \mathcal{N}(z; 0, 1)$, $\tilde{\boldsymbol{n}}$ is a noise vector whose components are i.i.d. from the standard normal distribution $\mathcal{N}(\tilde{\boldsymbol{n}}; \boldsymbol{0}_d, I_d)$, and $\tilde{\eta} \in \mathbb{R}$ is a scalar parameter to control the strength of the noise.

We also define the linear discriminator as

$$\psi(\boldsymbol{x}; \boldsymbol{v}) \triangleq f\left(\frac{1}{\sqrt{d}} \boldsymbol{v}^\top \boldsymbol{x}\right), \tag{9}$$

where $\boldsymbol{x}$ is an input vector, which can be either the real data $\boldsymbol{x}^\mu$ from Eq. (7) or the fake one $g(z^{\tilde{\mu}}; \boldsymbol{w})$ from Eq. (8). The vector $\boldsymbol{v} \in \mathbb{R}^d$ is a learnable parameter, and $f : \mathbb{R} \to \mathbb{R}$ can be any function.

**Training algorithm**   The GAN is trained by solving the following min-max optimization problem:

$$\min_{\boldsymbol{w} \in \mathbb{R}^d} \max_{\boldsymbol{v} \in \mathbb{R}^d} V(\boldsymbol{w}, \boldsymbol{v}; \mathcal{D}), \tag{10}$$

where

$$V(\boldsymbol{w}, \boldsymbol{v}; \mathcal{D}, \tilde{\mathcal{D}}) \triangleq \sum_{\mu=1}^{n} \phi\left(\psi(\boldsymbol{x}^\mu; \boldsymbol{v})\right) - \sum_{\tilde{\mu}=1}^{\tilde{n}} \tilde{\phi}\left(\psi(g(z^{\tilde{\mu}}; \boldsymbol{w}); \boldsymbol{v})\right) - \frac{\lambda}{2}\|\boldsymbol{v}\|^2 + \frac{\tilde{\lambda}}{2}\|\boldsymbol{w}\|^2, \tag{11}$$

and $\tilde{\mathcal{D}} = \{z^\mu\}_{\tilde{\mu}=1}^{\tilde{n}}$ are the latent values of the fake data. The last two terms are regularization terms, where $\lambda$ and $\tilde{\lambda}$ control the regularization strength. This value function defined in Eq. (11) is a general form that includes various types of GANs. Specifically, when $\phi = \tilde{\phi}$ and $\|\phi\|_L \leq 1$, it represents a Wasserstein GANs (WGANs) (Arjovsky et al., 2017) and, when $\phi(x) = \log \sigma(x)$ and $\tilde{\phi}(x) = -\log(1 - \sigma(x))$ with $\sigma$ being the sigmoid function, it corresponds to the Vanilla GANs, which minimize the JS-divergence (Goodfellow et al., 2014a). As we assumes a linear discriminator, $V(\boldsymbol{w}, \boldsymbol{v}; \mathcal{D}, \tilde{\mathcal{D}})$ can be expressed as a function of linear combinations $\boldsymbol{v}^\top \boldsymbol{x}^\mu / \sqrt{d}$ and $\boldsymbol{v}^\top g(z^{\tilde{\mu}}; \boldsymbol{w}) / \sqrt{d}$ as follows:

$$V(\boldsymbol{w}, \boldsymbol{v}; \mathcal{D}, \tilde{\mathcal{D}}) = \sum_{\mu=1}^{n} \phi\left(\frac{1}{\sqrt{d}} \boldsymbol{v}^\top \boldsymbol{x}^\mu\right) - \sum_{\tilde{\mu}=1}^{\tilde{n}} \tilde{\phi}\left(\frac{1}{\sqrt{d}} \boldsymbol{v}^\top g(z^{\tilde{\mu}}; \boldsymbol{w})\right) - \frac{\lambda}{2}\|\boldsymbol{v}\|^2 + \frac{\tilde{\lambda}}{2}\|\boldsymbol{w}\|^2, \tag{12}$$

where, for clarity in the subsequent analysis, we redefined $\phi$ and $\tilde{\phi}$ as functions of the linear combinations $\boldsymbol{v}^\top \boldsymbol{x}^\mu / \sqrt{d}$ and $\boldsymbol{v}^\top g(z^{\tilde{\mu}}; \boldsymbol{w}) / \sqrt{d}$.

**Generalization error**   In the ideal case where the generator perfectly learns the underlying true probability distribution, we have $\boldsymbol{w}^* = \boldsymbol{w}$. Therefore, we define the generalization error $\varepsilon_g$ as

$$\varepsilon_g(\bar{\boldsymbol{w}}, \boldsymbol{w}^*) \triangleq \frac{1}{d} \mathbb{E}_{\mathcal{D}}\left[\|\bar{\boldsymbol{w}} - \boldsymbol{w}^*\|^2\right], \tag{13}$$

where $\bar{\boldsymbol{w}}$ denotes the min-max optimal value in Eq. (10). The generalization error, $\varepsilon_g$, quantifies the accuracy of signal recovery from the training data.

## 5.2 Replica Calculation

We apply the replica formalism sketched in Section 3 to derive a set of deterministic equations characterizing the typical behavior of GANs.

In this problem setting, the replicated partition function $Z_\gamma$ in Eq. (6) can be expressed as

$$Z_\gamma(\beta_{\min}, \beta_{\max}) = \prod_{a=1}^{\gamma} \int_{\mathbb{R}^d} d\boldsymbol{w}^a \prod_{l=1}^{p} \int_{\mathbb{R}^d} d\boldsymbol{v}^{al} \left( \mathbb{E}_{c,\boldsymbol{n}} e^{\beta_{\max} \sum_{al} \phi\left( \frac{1}{d}(\boldsymbol{v}^{al})^\top \boldsymbol{w}^* c + \sqrt{\frac{\eta}{d}}(\boldsymbol{v}^{al})^\top \boldsymbol{n} \right)} \right)^n$$

$$\times \left( \mathbb{E}_z e^{-\beta_{\max} \sum_{al} \tilde{\phi}\left( \frac{1}{d}(\boldsymbol{v}^{al})^\top \boldsymbol{w}^a z + \sqrt{\frac{\tilde{\eta}}{d}}(\boldsymbol{v}^{al})^\top \tilde{\boldsymbol{n}} \right)} \right)^{\tilde{n}} e^{\frac{\beta_{\max}}{2} \sum_{al} \left( \tilde{\lambda} \|\boldsymbol{w}^a\|^2 - \lambda \|\boldsymbol{v}^{al}\|^2 \right)}.$$

To take the average over $\boldsymbol{n}$ and $\tilde{\boldsymbol{n}}$, we notice that since $\boldsymbol{n}$ and $\tilde{\boldsymbol{n}}$ follow a multivariate normal distribution $\mathcal{N}(\tilde{\boldsymbol{n}}; \boldsymbol{0}_d, I_d)$, the quantities $\boldsymbol{u} = ((\boldsymbol{v}^{al})^\top \boldsymbol{n}/\sqrt{d})_{a,l}$ and $\tilde{\boldsymbol{u}} = ((\boldsymbol{v}^{al})^\top \tilde{\boldsymbol{n}}/\sqrt{d})_{a,l}$ also follow a Gaussian multivariate distribution as

$$p(\boldsymbol{u}) = p(\tilde{\boldsymbol{u}}) = \mathcal{N}(\boldsymbol{0}_{\gamma p}, \boldsymbol{Q}),$$

where

$$\boldsymbol{Q} = (Q_{ls}^{ab}) \in \mathbb{R}^{\gamma p \times \gamma p}, \quad Q_{ls}^{ab} = \frac{1}{d}(\boldsymbol{v}^{al})^\top \boldsymbol{v}^{bs}.$$

To conduct further computations, we introduce auxiliary variables through the following identities:

$$1 = \prod_{abls} d \int \delta(dQ_{ls}^{ab} - (\boldsymbol{v}^{al})^\top \boldsymbol{v}^{bs}) dQ_{ls}^{ab} = \prod_{al} d \int \delta(dm_l^a - (\boldsymbol{v}^{al})^\top \boldsymbol{w}^*) dm_l^a = \prod_{al} d \int \delta(db_l^a - (\boldsymbol{v}^{al})^\top \boldsymbol{w}^a) db_l^a.$$

The replicated partition function can then be expressed as

$$Z_\gamma(\beta_{\min}, \beta_{\max}) = \int d\boldsymbol{Q} d\boldsymbol{m} d\boldsymbol{b} e^{\beta_{\min} d(\mathcal{S}(\boldsymbol{Q},\boldsymbol{m},\boldsymbol{b}) + \mathcal{T}(\boldsymbol{Q},\boldsymbol{m},\boldsymbol{b}))},$$

where the entropic term $\mathcal{S}(\boldsymbol{Q}, \boldsymbol{m}, \boldsymbol{b})$ and energetic term $\mathcal{T}(\boldsymbol{Q}, \boldsymbol{m}, \boldsymbol{b})$ are defined as follows:

$$\mathcal{S}(\boldsymbol{Q}, \boldsymbol{m}, \boldsymbol{b}) \triangleq \frac{1}{d\beta_{\min}} \ln \int \prod_{al} d\boldsymbol{w}^a d\boldsymbol{v}^{al} \prod_{abls} d \int \delta(dQ_{ls}^{ab} - (\boldsymbol{v}^{al})^\top \boldsymbol{v}^{bs})$$

$$\times \prod_{al} d \int \delta(dm_l^a - (\boldsymbol{v}^{al})^\top \boldsymbol{w}^*) d \int \delta(db_l^a - (\boldsymbol{v}^{al})^\top \boldsymbol{w}^a) e^{\frac{\beta_{\max}}{2} \sum_{al} \left( \tilde{\lambda} \|\boldsymbol{w}^a\|^2 - \lambda \|\boldsymbol{v}^{al}\|^2 \right)},$$

$$\mathcal{T}(\boldsymbol{Q}, \boldsymbol{m}, \boldsymbol{b}) \triangleq \frac{\alpha}{\beta_{\min}} \ln \left( \int Dc \int d\boldsymbol{u} p(\boldsymbol{u}) e^{\beta_{\max} \sum_{al} \phi\left( \frac{1}{d}(\boldsymbol{v}^{al})^\top \boldsymbol{w}^* c + \sqrt{\frac{\eta}{d}}(\boldsymbol{v}^{al})^\top \boldsymbol{n} \right)} \right)$$

$$+ \frac{\tilde{\alpha}}{\beta_{\min}} \ln \left( \int Dz \int d\tilde{\boldsymbol{u}} p(\tilde{\boldsymbol{u}}) e^{-\beta_{\max} \sum_{al} \tilde{\phi}\left( \frac{1}{d}(\boldsymbol{v}^{al})^\top \boldsymbol{w}^a z + \sqrt{\frac{\tilde{\eta}}{d}}(\boldsymbol{v}^{al})^\top \tilde{\boldsymbol{n}} \right)} \right).$$

Using the Fourier representation of the delta function, $\mathcal{S}(\boldsymbol{Q}, \boldsymbol{m}, \boldsymbol{b})$ is further expressed as

$$\mathcal{S}(\boldsymbol{Q}, \boldsymbol{m}, \boldsymbol{b}) = \frac{1}{d\beta_{\min}} \log \int d\tilde{\boldsymbol{Q}} d\tilde{\boldsymbol{m}} d\tilde{\boldsymbol{b}} e^{d\left( \frac{1}{2}\text{tr}\tilde{\boldsymbol{Q}}\boldsymbol{Q} - \tilde{\boldsymbol{m}}^\top \boldsymbol{m} - \tilde{\boldsymbol{b}}^\top \boldsymbol{b} \right)}$$

$$\left( \int \prod_{al} dw^a dv^{al} e^{-\frac{1}{2} \sum_{abls} \tilde{Q}_{ls}^{ab} v^{al} v^{bs} + w^* \sum_{al} \tilde{m}_l^a v^{al} + \sum_{al} \tilde{b}_l^a w^a v^{al} + \frac{\beta_{\max}}{2} \sum_{al} \left( \tilde{\lambda}(w^a)^2 - \lambda(v^{al})^2 \right)} \right)^d. \quad (14)$$

**Replica symmetric ansatz**   Here, we assume the following symmetric structure:

$$\forall a, b \in [\gamma], \forall l, s \in [p], \quad Q_{ls}^{ab} = q + \frac{\Delta}{\beta_{\min}} \delta_{ab} + \frac{\chi}{\beta_{\max}} \delta_{ls} \delta_{ab}, \quad (15)$$

$$\forall a, b \in [\gamma], \forall l, s \in [p], \quad \tilde{Q}_{ls}^{ab} = \beta_{\max} \hat{q} \delta_{ls} \delta_{ab} - \frac{\beta_{\max}^2}{\beta_{\min}} \hat{\Delta} \delta_{ab} - \beta_{\max}^2 \hat{\chi}, \quad (16)$$

$$\forall a \in [\gamma], l \in [p], \quad m_l^a = m, \quad \tilde{m}_l^a = \beta_{\max} \hat{m}, \quad (17)$$

$$\forall a \in [\gamma], l \in [p], \quad b_l^a = b, \quad \hat{b}_l^a = \beta_{\max} \hat{b}. \quad (18)$$

This replica symmetric (RS) structure restricts the integration of the replicated weight parameters $\{\boldsymbol{w}^a\}$, $\{\boldsymbol{v}^{al}\}$ across the entire $\mathbb{R}^{(d\times\gamma p)}\times\mathbb{R}^{(d\times\gamma p)}$ to a subspace that satisfies the constraints in Eq. (15)–(18). This structure, along with scaling by the maximum and minimum beta values, is similar to the standard one-step replica symmetry breaking (1RSB) (Mézard et al., 1987; Takahashi & Kabashima, 2022).

We now turn to the entropic term $\mathcal{S}(\boldsymbol{Q},\boldsymbol{m},\boldsymbol{b})$. The terms that exclude the integrals with respect to $\{\boldsymbol{v}^{al}\}$ and $\{\boldsymbol{w}^a\}$ can be expressed as

$$
\frac{1}{2}\mathrm{tr}\tilde{\boldsymbol{Q}}\boldsymbol{Q} - \tilde{\boldsymbol{m}}^\top\boldsymbol{m} - \tilde{\boldsymbol{b}}^\top\boldsymbol{b}
$$
$$
= \gamma\beta_{\min}\left(-\frac{1}{2}\left(\hat{q}\left(q+\frac{\Delta}{\beta_{\min}}+\frac{\chi}{\beta_{\max}}\right)-\chi\left(\hat{\chi}+\frac{\hat{\Delta}}{\beta_{\min}}\right)+\hat{\chi}\Delta+\hat{\Delta}q+\frac{\Delta\hat{\Delta}}{\beta_{\min}}\right)+\hat{m}m+\hat{b}b\right). \quad (19)
$$

The term that includes the integrals with respect to $\{\boldsymbol{v}^{al}\}$ and $\{\boldsymbol{w}^a\}$ can be expressed as

$$
\mathbb{E}_z\left[\int\prod_{al}dw^a D\zeta^a dv^{al} e^{-\frac{1}{2}\beta_{\max}(\hat{q}+\lambda)\sum_{al}(v^{al})^2+\beta_{\max}\sum_{al}\left(\sqrt{\frac{\hat{\Delta}}{\beta_{\min}}}\zeta^a+\sqrt{\hat{\chi}}z+w^*\hat{m}+w^a\hat{b}\right)v^{al}-\frac{\tilde{\lambda}\beta_{\min}}{2}\sum_a(w^a)^2}\right],
$$
$$
= \mathbb{E}_z\left[\int\prod_a dw^a D\zeta^a\left(\int dv^a e^{-\frac{1}{2}\beta_{\max}(\hat{q}+\lambda)(v^a)^2+\beta_{\max}\left(\sqrt{\frac{\hat{\Delta}}{\beta_{\min}}}\zeta^a+\sqrt{\hat{\chi}}z+w^*\hat{m}+w^a\hat{b}\right)v^a}\right)^p e^{-\frac{\tilde{\lambda}\beta_{\min}}{2}\sum_a(w^a)^2}\right],
$$
$$
= \mathbb{E}_z\left[\int\prod_a dw^a D\zeta^a e^{-\frac{\tilde{\lambda}\beta_{\min}}{2}\sum_a(w^a)^2-\frac{\beta_{\min}}{2(\hat{q}+\lambda)}\sum_a\left(\sqrt{\frac{\hat{\Delta}}{\beta_{\min}}}\zeta^a+\sqrt{\hat{\chi}}z+w^*\hat{m}+w^a\hat{b}\right)^2}\right],
$$
$$
= \mathbb{E}_z\left[\left(\int dwd\zeta e^{\beta_{\min}\left(-\frac{1}{2}\zeta^2-\frac{\tilde{\lambda}}{2}w^2-\frac{(\sqrt{\hat{\Delta}}\zeta+\sqrt{\hat{\chi}}z+w^*\hat{m}+w\hat{b})^2}{2(\hat{q}+\lambda)}\right)}\right)^\gamma\right].
$$

This can be derived using the identity, for any $a\in\mathbb{R}_+$ and any $x\in\mathbb{R}$, $e^{\frac{a}{2}x^2}=\int Dze^{\sqrt{a}zx}$. Summarizing these results, the entropic term can be written as

$$
\mathcal{S}(\boldsymbol{Q},\boldsymbol{m},\boldsymbol{b},\tilde{\boldsymbol{Q}},\tilde{\boldsymbol{m}},\tilde{\boldsymbol{b}}) = \gamma\left(-\frac{1}{2}\left(\hat{q}\left(q+\frac{\Delta}{\beta_{\min}}+\frac{\chi}{\beta_{\max}}\right)-\chi\left(\hat{\chi}+\frac{\hat{\Delta}}{\beta_{\min}}\right)+\hat{\chi}\Delta+\hat{\Delta}q+\frac{\Delta\hat{\Delta}}{\beta_{\min}}\right)+\hat{m}m+\hat{b}b\right.
$$
$$
\left.+\frac{1}{\beta_{\min}}\int Dz\log\int dwd\zeta e^{\beta_{\min}\left(-\frac{1}{2}\zeta^2-\frac{\tilde{\lambda}}{2}w^2-\frac{(\sqrt{\hat{\Delta}}\zeta+\sqrt{\hat{\chi}}z+w^*\hat{m}+w\hat{b})^2}{2(\hat{q}+\lambda)}\right)}\right).
$$

By taking the limit as $\beta_{\max}\to\infty$ followed by $\beta_{\min}\to\infty$, we obtain

$$
\mathcal{S}(\boldsymbol{Q},\boldsymbol{m},\boldsymbol{b},\tilde{\boldsymbol{Q}},\tilde{\boldsymbol{m}},\tilde{\boldsymbol{b}}) = -\frac{\gamma}{2}\left(q(\hat{q}+\hat{\Delta})-(\chi-\Delta)\hat{\chi}-2m\hat{m}-2b\hat{b}+\frac{\tilde{\lambda}(\hat{m}^2+\hat{\chi})}{\hat{b}^2+(\hat{q}+\hat{\Delta}+\lambda)\tilde{\lambda}}\right).
$$

We next turn to the energetic term $\mathcal{T}(\boldsymbol{Q},\boldsymbol{m},\boldsymbol{b})$. Under the RS ansatz, $\boldsymbol{u}$ follows

$$
u^{al} = \sqrt{\frac{\chi}{\beta_{\max}}}x^{al}+\sqrt{\frac{\Delta}{\beta_{\min}}}y^a+\sqrt{q}\xi,
$$

where $\tilde{x}^{al}$, $x^{al}$, $\tilde{y}^{al}$, $y^{al}$, $\xi$, and $\tilde{\xi}$ follow the standard normal distribution $\mathcal{N}(\tilde{\xi}; 0, 1)$. Then, the energetic term $\mathcal{T}(\boldsymbol{Q}, \boldsymbol{m}, \boldsymbol{b})$ can be expand as

$$
\mathcal{T}(\boldsymbol{Q}, \boldsymbol{m}, \boldsymbol{b}) \triangleq \underbrace{\frac{\alpha}{\beta_{\min}} \ln \left( \int Dc \int d\boldsymbol{u} p(\boldsymbol{u}) e^{\beta_{\max} \sum_{al} \phi\left( \frac{1}{d}(\boldsymbol{v}^{al})^\top \boldsymbol{w}^* c + \sqrt{\frac{\eta}{d}}(\boldsymbol{v}^{al})^\top \boldsymbol{n} \right)} \right)}_{(a)}
$$
$$
+ \underbrace{\frac{\tilde{\alpha}}{\beta_{\min}} \ln \left( \int Dz \int d\tilde{\boldsymbol{u}} p(\tilde{\boldsymbol{u}}) e^{-\beta_{\max} \sum_{al} \tilde{\phi}\left( \frac{1}{d}(\boldsymbol{v}^{al})^\top \boldsymbol{w}^a z + \sqrt{\frac{\tilde{\eta}}{d}}(\boldsymbol{v}^{al})^\top \tilde{\boldsymbol{n}} \right)} \right)}_{(b)}.
$$

The term (a) can be simplified as

$$
(a) = \frac{\alpha}{\beta_{\min}} \ln \mathbb{E}_{c,\xi} \left[ \int \prod_{al} Dy^a Dx^{al} e^{\beta_{\max} \sum_{al} \phi\left( mc + \sqrt{\eta}\left( \sqrt{\frac{\chi}{\beta_{\max}}} x^{al} + \sqrt{\frac{\Delta}{\beta_{\min}}} y^a + \sqrt{q}\xi \right) \right)} \right],
$$
$$
= \frac{\alpha}{\beta_{\min}} \ln \mathbb{E}_{c,\xi} \left[ \left( \int Dy \left( \int Dx e^{\beta_{\max} \phi\left( mc + \sqrt{\eta}\left( \sqrt{\frac{\chi}{\beta_{\max}}} x + \sqrt{\frac{\Delta}{\beta_{\min}}} y + \sqrt{q}\xi \right) \right)} \right)^p \right)^\gamma \right],
$$
$$
= \frac{\alpha}{\beta_{\min}} \gamma \mathbb{E}_{c,\xi} \left[ \log \int Dy \left( \int Dx e^{\beta_{\max} \phi\left( mc + \sqrt{\eta}\left( \sqrt{\frac{\chi}{\beta_{\max}}} x + \sqrt{\frac{\Delta}{\beta_{\min}}} y + \sqrt{q}\xi \right) \right)} \right)^p \right] + \mathcal{O}_\gamma(\gamma^2),
$$
$$
= \frac{\alpha}{\beta_{\min}} \gamma \mathbb{E}_{c,\xi} \left[ \log \int dy e^{-\frac{\beta_{\min}}{2} y^2} \left( \int dx e^{-\frac{\beta_{\max}}{2} x^2 + \beta_{\max} \phi\left( mc + \sqrt{\eta}\left( \sqrt{\chi} x + \sqrt{\Delta} y + \sqrt{q}\xi \right) \right)} \right)^p \right] + \mathcal{O}_\gamma(\gamma^2).
$$

Taking the limit as $\beta_{\max} \to \infty$ followed by $\beta_{\min} \to \infty$, we obtain:

$$
(a) = \alpha\gamma \mathbb{E}_{c,\xi} \left[ \max_y \left\{ -\frac{1}{2} y^2 - \max_x \left\{ -\frac{1}{2} x^2 + \phi\left( mc + \sqrt{\eta}\left( \sqrt{\chi} x + \sqrt{\Delta} y + \sqrt{q}\xi \right) \right) \right\} \right\} \right].
$$

Similarly, the term (b) is also expressed as

$$
(b) = \frac{\tilde{\alpha}}{\beta_{\min}\gamma} \mathbb{E}_{z,\tilde{\xi}} \left[ \ln \int d\tilde{y} e^{-\frac{\beta_{\min}}{2} y^2} \left( \int d\tilde{x} e^{-\frac{\beta_{\max}}{2} \tilde{x}^2 - \beta_{\max} \tilde{\phi}\left( bz + \sqrt{\tilde{\eta}}\left( \sqrt{\chi} \tilde{x} + \sqrt{\Delta} \tilde{y} + \sqrt{q}\tilde{\xi} \right) \right)} \right)^p \right] + \mathcal{O}_\gamma(\gamma^2).
$$

Taking the same limits, we find:

$$
(b) = \tilde{\alpha}\gamma \mathbb{E}_{z,\tilde{\xi}} \left[ \max_{\tilde{y}} \left\{ -\frac{1}{2} y^2 - \max_{\tilde{x}} \left\{ -\frac{1}{2} \tilde{x}^2 - \tilde{\phi}\left( bz + \sqrt{\tilde{\eta}}\left( \sqrt{\chi} \tilde{x} + \sqrt{\Delta} \tilde{y} + \sqrt{q}\tilde{\xi} \right) \right) \right\} \right\} \right].
$$

Putting the entropic term and energetic term together, the free energy density is given by

$$
f = \underset{\substack{\hat{q},\hat{\chi},\hat{m},\hat{b} \\ q,\delta,\chi,m,b}}{\text{extr}} \frac{1}{2} \left( q\hat{q} - (\chi - \Delta)\hat{\chi} - 2(m\hat{m} + b\hat{b}) + \frac{\tilde{\lambda}(\hat{m}^2 + \hat{\chi})}{\hat{b}^2 + (\hat{q} + \lambda)\tilde{\lambda}} - 2(\alpha\Phi(q, \Delta, \chi, m, b) + \tilde{\alpha}\tilde{\Phi}(q, \Delta, \chi, m, b)) \right),
$$

where

$$
\Phi(q, \Delta, \chi, m, b) = \mathbb{E}_{c,\xi} \left[ \max_y \left\{ -\frac{1}{2} y^2 - \max_x \left\{ -\frac{1}{2} x^2 + \phi\left( mc + \sqrt{\eta}\left( \sqrt{\chi} x + \sqrt{\Delta} y + \sqrt{q}\xi \right) \right) \right\} \right\} \right], \quad (20)
$$

$$
\tilde{\Phi}(q, \Delta, \chi, m, b) = \mathbb{E}_{z,\tilde{\xi}} \left[ \max_{\tilde{y}} \left\{ -\frac{1}{2} \tilde{y}^2 - \max_{\tilde{x}} \left\{ -\frac{1}{2} \tilde{x}^2 - \tilde{\phi}\left( bz + \sqrt{\tilde{\eta}}\left( \sqrt{\chi} \tilde{x} + \sqrt{\Delta} \tilde{y} + \sqrt{q}\tilde{\xi} \right) \right) \right\} \right\} \right]. \quad (21)
$$

Note that the min and max operations are involved in the two-level optimization described in Eqs. (20) and (21).

## 5.3 Results: Application to Simple GANs

In this subsection, following Wang et al. (2019), we apply the formulation derived above to the simple WGAN to demonstrate its generalization properties and conduct a sensitivity analysis of the ratio $r$ fake to real data.

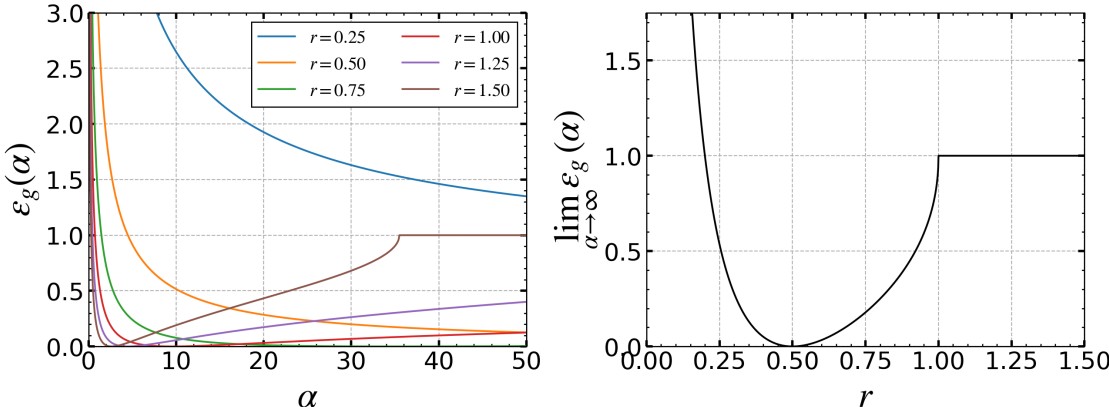

Figure 1: (Left) Generalization error as a function of sample complexity $\alpha$ for different values of the ratio $r$. (Right) Asymptotic generalization error $\lim_{\alpha \to \infty} \varepsilon(\alpha)$ as a function of the the ratio $r$.

**Self-consistent Equations** We consider the case where the functions $\phi(x)$ and $\tilde{\phi}(x)$ are both quadratic, defined as $\phi(x) = \tilde{\phi}(x) = x^2/2$. This setting allows for an explicit calculation of the free energy density, which is given by

$$f = \operatorname*{extr}_{\substack{q,\chi,m,b \\ \hat{q},\hat{\chi},\hat{m},\hat{b}}} \left[ \frac{1}{2} \left( q\hat{q} - \chi\hat{\chi} - 2m\hat{m} - 2b\hat{b} + \frac{\tilde{\lambda}(\hat{m}^2 + \hat{\chi})}{\hat{b}^2 + (\hat{q} + \lambda)\tilde{\lambda}} - \frac{\alpha(\eta q + m^2)}{\eta\chi - 1} - \frac{\tilde{\alpha}(\tilde{\eta}q + b^2)}{\tilde{\eta}\chi + 1} \right) \right]. \tag{22}$$

To find the extremum in Eq. (22), we require that the gradient with respect to each order parameter equals zero. This results in the following set of self-consistent equations:

$$q = \frac{\tilde{\lambda}^2(\hat{m}^2 + \hat{\chi})}{(\hat{b}^2 + \tilde{\lambda}(\hat{q} + \lambda))^2}, \quad \chi = \frac{\tilde{\lambda}}{\hat{b}^2 + \tilde{\lambda}(\hat{q} + \lambda)}, \quad m = \frac{\hat{m}\tilde{\lambda}}{\hat{b}^2 + \tilde{\lambda}(\hat{q} + \lambda)}, \quad b = -\frac{\hat{b}\tilde{\lambda}(\hat{m}^2 + \hat{\chi})}{(\hat{b}^2 + \tilde{\lambda}(\hat{q} + \lambda))^2},$$

$$\hat{q} = \frac{\alpha\eta}{\eta\chi - 1} + \frac{\tilde{\alpha}\tilde{\eta}}{\tilde{\eta}\chi + 1}, \quad \hat{\chi} = \frac{\alpha\eta(q\eta + m^2\rho)}{(\eta\chi - 1)^2} + \frac{\tilde{\alpha}\tilde{\eta}(q\tilde{\eta} + d^2\rho)}{(\tilde{\eta}\chi + 1)^2}, \quad \hat{m} = \frac{\alpha m}{\eta\chi - 1}, \quad \hat{b} = -\frac{\tilde{\alpha}b}{\tilde{\eta}\chi + 1}.$$

**Learning Curve** For simplicity, we set $\tilde{\alpha} = r\alpha$ and $\lambda = \tilde{\lambda} = \eta = \tilde{\eta} = 1$. Our analysis focuses on how the generalization error depends on $\alpha$ while varying the ratio $r$, as generating fake data from the generator is generally much easier than collecting real data. Fig. 1 (Left) shows the dependence of generalization error on sample complexity $\alpha$ for various values of the ratio $r$. The results demonstrate a sharp decline in the generalization error as the ratio $r$ increases. However, when $r$ becomes large, the generalization error increases in the region where $\alpha$ is large, eventually leading to a phase where no learning occurs, and the generalization error equals 1. This implies that as $\alpha$ increases, the learning becomes dominated by only fake data.

In contrast, for smaller $r$, real data consistently dominates the objective function $V(\boldsymbol{w}, \boldsymbol{v}; \mathcal{D}, \tilde{\mathcal{D}})$, resulting in a steady decrease in generalization error. However, the reduced influence of the fake data component in the objective function, which drives the learning of the generator, requires a significantly larger amount of real data for effective generator training.

**Asymptotic Generalization Error** We next analyze the asymptotic behavior of the generalization error when the sample complexity $\alpha$ becomes sufficiently large. The asymptotic behavior of the generalization error as a function of $\alpha$ is given by

$$\varepsilon_g = \begin{cases} \frac{1 - 2\sqrt{\frac{1-r}{r}}r}{r} + \frac{2\sqrt{2}\left(\sqrt{\frac{1-r}{r}}r + r - 1\right)}{(r-1)r\alpha^{1/2}} + \mathcal{O}_\alpha(\alpha^{-1}) & r \leq 1, \\ 1 + \mathcal{O}_\alpha(\alpha^{-1}) & r > 1. \end{cases}$$

The results for $\alpha \to \infty$ are shown in Fig. 1(Right). The optimal ratio is $r = 1/2$, indicating that using fake data approximately equal to half of the real data is effective when the dataset approaches infinity. At $r = 1$, a phase transition occurs, suggesting that the model changes from a phase of effective learning phase to one where fake data becomes dominant. Beyond this point, for $r \geq 1$, the model fails to learn any meaningful signal $\boldsymbol{w}^*$, and the generalization error is 1.

Furthermore, when $r = 1/2$, the generalization error scales as $\varepsilon_g \sim \alpha^{-1}$, which represents the optimal asymptotic behavior for a model-matched scenario. These results demonstrate the critical role of the ratio $r$ in determining learning performance. Therefore, tuning the ratio $r$ according to the available real data is crucial for achieving optimal performance. In practice, it is known that in training GANs, the stability of learning can deteriorate depending on the ratio $r$ of fake to real data. This theoretical analysis provides insights into the importance of the ratio $r$ and is expected to contribute to improving learning algorithms.

## 6 Conclusion

This study introduces a statistical mechanical formalism to analyze high-dimensional min-max optimization problems, focusing on the critical order of min and max operations in non-convex scenarios. Our goal was to perform a sensitivity analysis of equilibrium values, providing new insights into their properties and generalization performance.

We applied this approach to a simple min-max game, evaluated the generalization performance of GANs, and derived the optimal ratio of fake to real data for effective learning. This successful application not only validates the approach but also opens the way for extending this formalism to more complex min-max problems and broader applications, suggesting a promising direction for significant advancements in machine learning and optimization.

### Acknowledgments

We thank T. Takahashi, K. Okajima, Y. Nagano, and K. Nakaishi for useful discussions and suggestions. This work was supported by JST Grant Number JPMJPF2221 and JPSJ Grant-in-Aid for Scientific Research Number 23H01095. Additionally, YI was supported by the WINGS-FMSP program at the University of Tokyo.

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

## A  Derivation of Theorem 4.1 proof

In this section, we provide the derivation proof of Theorem 4.1. The derivation begins with the calculation of the free energy density without the analytic continuation of $p = -\beta_{\min}/\beta_{\max}$ to $p \in \mathbb{N}$. The free energy density in Eq. (2) is connected to the effective Hamiltonian, $\mathcal{H}_{\mathrm{eff}}(\boldsymbol{x}; \boldsymbol{W})$, which is defined through the relationship:

$$f(\beta_{\min}, \beta_{\max}; \boldsymbol{W}) = -\frac{1}{\beta_{\min} d_x} \mathbb{E}_{\boldsymbol{W}} \log \sum_{\boldsymbol{x}} \exp\left(-\beta_{\min} \mathcal{H}_{\mathrm{eff}}(\boldsymbol{x}; \boldsymbol{W})\right).$$

The effective Hamiltonian is given by

$$\mathcal{H}_{\mathrm{eff}}(\boldsymbol{x}; \boldsymbol{W})$$
$$= \frac{1}{\beta_{\max}} \log \sum_{\boldsymbol{y}} e^{\beta_{\max} V(\boldsymbol{x}, \boldsymbol{y}; \boldsymbol{W})},$$
$$= \frac{1}{\beta_{\max}} \log \sum_{\boldsymbol{y}} e^{\beta_{\max}\left(\frac{w_{xx} d_x}{2}\left(\frac{\boldsymbol{x}^\top \mathbf{1}_{d_x}}{d_x}\right)^2 + \frac{d_y w_{yy}}{2}\left(\frac{\boldsymbol{y}^\top \mathbf{1}_{d_y}}{d_y}\right)^2 + w_{xy} \sqrt{d_x d_y}\left(\frac{\boldsymbol{x}^\top \mathbf{1}_{d_x}}{d_x}\right)\left(\frac{\boldsymbol{y}^\top \mathbf{1}_{d_y}}{d_y}\right)\right)}$$
$$\times e^{\beta_{\max}\left(b_x d_x\left(\frac{\boldsymbol{x}^\top \mathbf{1}_{d_x}}{d_x}\right) + b_y d_y\left(\frac{\boldsymbol{y}^\top \mathbf{1}_{d_y}}{d_y}\right)\right)},$$
$$= \frac{1}{\beta_{\max}} \log e^{\beta_{\max} d_x\left(\frac{w_{xx}}{2}\left(\frac{\boldsymbol{x}^\top \mathbf{1}_{d_x}}{d_x}\right)^2 + b_x\left(\frac{\boldsymbol{x}^\top \mathbf{1}_{d_x}}{d_x}\right)\right)}$$
$$\times \sum_{\boldsymbol{y}} e^{\beta_{\max} d_y\left(\frac{w_{yy}}{2}\left(\frac{\boldsymbol{y}^\top \mathbf{1}_{d_y}}{d_y}\right)^2 + w_{xy}\sqrt{\frac{d_x}{d_y}}\left(\frac{\boldsymbol{x}^\top \mathbf{1}_{d_x}}{d_x}\right)\left(\frac{\boldsymbol{y}^\top \mathbf{1}_{d_y}}{d_y}\right) + b_y\left(\frac{\boldsymbol{y}^\top \mathbf{1}_{d_y}}{d_y}\right)\right)},$$
$$= \beta_{\max} d_x\left(\frac{w_{xx} d_x}{2}\left(\frac{\boldsymbol{x}^\top \mathbf{1}_{d_x}}{d_x}\right)^2 + b_x d_x\left(\frac{\boldsymbol{x}^\top \mathbf{1}_{d_x}}{d_x}\right)\right)$$
$$\times \frac{1}{\beta_{\max}} \log \int d\hat{m}^y dm^y e^{\beta_{\max} d_y\left(\frac{w_{yy}}{2}(m^y)^2 + w_{xy}\sqrt{\frac{d_x}{d_y}}\left(\frac{\boldsymbol{x}^\top \mathbf{1}_{d_x}}{d_x}\right)m^y + b_y m^y - \frac{1}{\beta_{\max}} m^y \hat{m}^y + \frac{1}{\beta_{\max}} \mathrm{Softplus}(\hat{m}^y)\right) + o(d_y)},$$

where $\mathrm{Softplus}(x) = \log(1 + e^x)$. To evaluate the integral with respect to $\hat{m}^y$ and $m^y$, we take the limit as $d_y \to \infty$ and apply the saddle point approximation. The effective Hamiltonian can be expressed as follows:

$$\mathcal{H}_{\mathrm{eff}}(\boldsymbol{x}; \boldsymbol{W}) = d_x\left(\frac{w_{xx}}{2}\left(\frac{\boldsymbol{x}^\top \mathbf{1}_{d_x}}{d_x}\right)^2 + b_x\left(\frac{\boldsymbol{x}^\top \mathbf{1}_{d_x}}{d_x}\right)\right.$$
$$\left. + \kappa \operatorname*{extr}_{m^y, \hat{m}^y}\left[\frac{w_{yy}}{2}(m^y)^2 + w_{xy}\kappa^{-1/2}\left(\frac{\boldsymbol{x}^\top \mathbf{1}_{d_x}}{d_x}\right)m^y + b_y m^y - \frac{1}{\beta_{\max}} m^y \hat{m}^y + \frac{1}{\beta_{\max}} \mathrm{Softplus}(\hat{m}^y)\right]\right).$$

where $\kappa = d_y/d_x$. By summing over $\boldsymbol{x}$, the free energy density can be calculated as follows:

$$f(\beta_{\min}, \beta_{\max}; \boldsymbol{W})$$

$$= -\frac{1}{\beta_{\min} d_x} \log \sum_{\boldsymbol{x}} e^{-\beta_{\min} d_x \left( \frac{w_{xx}}{2} \left( \frac{\boldsymbol{x}^\top \mathbf{1}_{d_x}}{d_x} \right)^2 + b_x \left( \frac{\boldsymbol{x}^\top \mathbf{1}_{d_x}}{d_x} \right) \right)}$$

$$\times e^{-\beta_{\min} d_x \left( \kappa \operatorname*{extr}_{m^y, \hat{m}^y} \left[ \frac{w_{yy}}{2}(m^y)^2 + w_{xy}\kappa^{-1/2} \left( \frac{\boldsymbol{x}^\top \mathbf{1}_{d_x}}{d_x} \right) m^y + b_y m^y - \frac{1}{\beta_{\max}} m^y \hat{m}^y + \frac{1}{\beta_{\max}} \operatorname{Softplus}(\hat{m}^y) \right] \right)},$$

$$= -\frac{1}{\beta_{\min} d_x} \log \left( \frac{d_x}{2\pi} \right) \int dm^x d\hat{m}^x e^{-\beta_{\min} d_x \left( \frac{w_{xx}}{2}(m^x)^2 + b_x m^x + \frac{1}{\beta_{\min}} m^x \hat{m}^x - \frac{1}{\beta_{\min}} \operatorname{Softplus}(\hat{m}^x) \right)}$$

$$\times e^{-\beta_{\min} d_x \left( \kappa \operatorname*{extr}_{m^y, \hat{m}^y} \left[ \frac{w_{yy}}{2}(m^y)^2 + w_{xy}\kappa^{-1/2} m^x m^y + b_y m^y - \frac{1}{\beta_{\max}} m^y \hat{m}^y + \frac{1}{\beta_{\max}} \operatorname{Softplus}(\hat{m}^y) \right] \right)},$$

$$= \operatorname*{extr}_{m^x, \hat{m}^x, m^y, \hat{m}^y} \left( \frac{w_{xx}}{2}(m^x)^2 + b_x m^x + \frac{1}{\beta_{\min}} m^x \hat{m}^x - \frac{1}{\beta_{\min}} \operatorname{Softplus}(\hat{m}^x) \right.$$

$$\left. + \frac{\kappa w_{yy}}{2}(m^y)^2 + w_{xy}\kappa^{1/2} m^x m^y + \kappa b_y m^y - \frac{\kappa}{\beta_{\max}} m^y \hat{m}^y + \frac{\kappa}{\beta_{\max}} \operatorname{Softplus}(\hat{m}^y) \right).$$

The final equality is obtained by applying the saddle point method to evaluate the integral. From the saddle point equations, the following expressions are

$$m^x = \sigma(\hat{m}^x), \quad m^y = \sigma(\hat{m}^y)$$

Further transformation of the equation yields the following expression:

$$f(\beta_{\min}, \beta_{\max}; \boldsymbol{W}) = \operatorname*{extr}_{m^x, m^y} \left[ \frac{w_{xx}}{2}(m^x)^2 + \frac{\kappa w_{yy}}{2}(m^y)^2 + w_{xy}\kappa^{1/2} m^x m^y + b_x m^x + \kappa b_y m^y \right.$$

$$\left. - \frac{1}{\beta_{\min}} H(m^x) + \frac{\kappa}{\beta_{\max}} H(m^y) \right], \quad (23)$$

where $H(x) = -x \log x - (1-x)\log(1-x)$ represents the binary cross-entropy.

Next, we proceed to evaluate the free energy density under analytic continuation in the replica method, which is expressed as

$$\hat{f}(\beta_{\min}, \beta_{\max}; \boldsymbol{W}) = -\frac{1}{\beta_{\min} d_x} \log \sum_{\boldsymbol{x}} \sum_{\boldsymbol{y}_1, \ldots, \boldsymbol{y}_p} e^{\beta_{\max} d_x \left( \frac{w_{xx}p}{2} \left( \frac{\boldsymbol{x}^\top \mathbf{1}_{d_x}}{d_x} \right)^2 + \frac{\kappa w_{yy}}{2} \sum_l \left( \frac{\boldsymbol{y}_l^\top \mathbf{1}_{d_y}}{d_y} \right)^2 \right)}$$

$$\times e^{\beta_{\max} d_x \left( w_{xy}\kappa^{1/2} \left( \frac{\boldsymbol{x}^\top \mathbf{1}_{d_x}}{d_x} \right) \sum_l \left( \frac{\boldsymbol{y}_l^\top \mathbf{1}_{d_y}}{d_y} \right) + b_x p \left( \frac{\boldsymbol{x}^\top \mathbf{1}_{d_x}}{d_x} \right) + \kappa b_y \sum_l \left( \frac{\boldsymbol{y}_l^\top \mathbf{1}_{d_y}}{d_y} \right) \right)}.$$

We introduce the order parameter through the Fourier transform representation of the delta function:

$$\hat{f}(\beta_{\min}, \beta_{\max}; \boldsymbol{W})$$

$$= -\frac{1}{\beta_{\min} d_x} \log \left( \frac{d_x d_y^p}{(2\pi)^{p+1}} \right) \int dm^x d\hat{m}^x \prod_l dm_l^y d\hat{m}_l^y e^{\beta_{\max} d_x \left( \frac{w_{xx}p}{2}(m^x)^2 + \frac{\kappa w_{yy}}{2} \sum_l (m_l^y)^2 + w_{xy}\kappa^{1/2} m^x \sum_l m_l^y \right)}$$

$$\times e^{\beta_{\max} d_x \left( b_x p m^x + \kappa b_y \sum_l m_l^y - \frac{1}{\beta_{\max}} \hat{m}^x m^x - \frac{\kappa}{\beta_{\max}} \sum_l \hat{m}_l^y m_l^y \right)} \sum_{\boldsymbol{x}} e^{\hat{m}^x \sum_i x_i} \sum_{\boldsymbol{y}_1, \ldots, \boldsymbol{y}_p} e^{\sum_l \hat{m}_l^y \sum_j y_j},$$

$$= -\frac{1}{\beta_{\min} d_x} \log \left( \frac{d_x d_y^p}{(2\pi)^{p+1}} \right) \int dm^x d\hat{m}^x \prod_l dm_l^y d\hat{m}_l^y e^{\beta_{\max} d_x \left( \frac{w_{xx}p}{2}(m^x)^2 + \frac{\kappa w_{yy}}{2} \sum_l (m_l^y)^2 + w_{xy}\kappa^{1/2} m^x \sum_l m_l^y \right)}$$

$$\times e^{\beta_{\max} d_x \left( b_x p m^x + \kappa b_y \sum_l m_l^y - \frac{1}{\beta_{\max}} \hat{m}^x m^x - \frac{\kappa}{\beta_{\max}} \sum_l \hat{m}_l^y m_l^y + \frac{1}{\beta_{\max}} \operatorname{Softplus}(\hat{m}^x) + \frac{\kappa}{\beta_{\max}} \sum_l \operatorname{Softplus}(\hat{m}_l^y) \right)}.$$

Under the assumption of replica symmetry, where $\forall l \in [p], \quad \hat{m}_l^y = m^y, m_l^y = m^y$, we can further reformulate the expression as follows:

$$
\begin{aligned}
&\hat{f}(\beta_{\min}, \beta_{\max}; \boldsymbol{W}) \\
&= -\frac{1}{\beta_{\min} d_x} \log \int dm^x d\hat{m}^x dm^y d\hat{m}^y e^{-\beta_{\min} d_x \left( \frac{w_{xx}}{2}(m^x)^2 + \frac{\kappa w_{yy}}{2}(m^y)^2 + w_{xy}\kappa^{1/2}m^x m^y \right)} \\
&\qquad \times e^{-\beta_{\min} d_x \left( b_x m^x + \kappa b_y m^y - \frac{1}{\beta_{\max} p}\hat{m}^x m^x - \frac{\kappa}{\beta_{\max}}\hat{m}^y m^y + \frac{1}{\beta_{\max} p}\mathrm{Softplus}(\hat{m}^x) + \frac{\kappa}{\beta_{\max}}\mathrm{Softplus}(\hat{m}^y) + o(d^x) + o(d^y) \right)}, \\
&= \operatorname*{extr}_{m^x, m^y, \hat{m}^x, \hat{m}^y} \left[ \frac{w_{xx}}{2}(m^x)^2 + \frac{\kappa w_{yy}}{2}(m^y)^2 + w_{xy}\kappa^{1/2}m^x m^y + b_x m^x + \kappa b_y m^y \right. \\
&\qquad\qquad\qquad \left. - \frac{1}{\beta_{\max} p}\hat{m}^x m^x - \frac{\kappa}{\beta_{\max}}\hat{m}^y m^y + \frac{1}{\beta_{\max} p}\mathrm{Softplus}(\hat{m}^x) + \frac{\kappa}{\beta_{\max}}\mathrm{Softplus}(\hat{m}^y) \right].
\end{aligned}
$$

The final equality is derived by handling the integral using the saddle point method. Consequently, the following saddle point equation is obtained:

$$
m^x = \sigma(\hat{m}^x), \quad m^y = \sigma(\hat{m}^y).
$$

Substituting these results, the following expression for the free energy density is derived as

$$
\begin{aligned}
\hat{f}(\beta_{\min}, \beta_{\max}; \boldsymbol{W}) = \operatorname*{extr}_{m^x, m^y} \left[ \frac{w_{xx}}{2}(m^x)^2 + \frac{\kappa w_{yy}}{2}(m^y)^2 + w_{xy}\kappa^{1/2}m^x m^y + b_x m^x + \kappa b_y m^y \right. \\
\left. - \frac{1}{\beta_{\min}}H(m^x) + \frac{\kappa}{\beta_{\max}}H(m^y) \right]. \quad (24)
\end{aligned}
$$

This result coincides with the exact free energy density $f(\beta_{\min}, \beta_{\max}; \boldsymbol{W})$, derived without the need for analytic continuation.

## B  Generalization of Theorem 4.1 to Finite Eigenmodes

In this section, we generalize Theorem 4.1 to cases where $W_{xx}$, $W_{xy}$, and $W_{yy}$ are decomposed into a finite number of eigenmodes as follows:

$$
W_{xx} = \sum_{l=1}^{L} \alpha_l \boldsymbol{a}_l \boldsymbol{a}_l^\top, \ W_{yy} = \sum_{m=1}^{M} \beta_m \boldsymbol{b}_m \boldsymbol{b}_m^\top, \ W_{xy} = \sum_{n=1}^{N} \gamma_n \boldsymbol{c}_n \boldsymbol{c}_n^\top, \ \alpha_l, \beta_m, \gamma_n, L, M, N = \mathcal{O}_{d_x, d_y}(1).
$$

In this formulation, $V(\boldsymbol{x}, \boldsymbol{y}; \boldsymbol{W})$ can be expanded as follows:

$$
\begin{aligned}
V(\boldsymbol{x}, \boldsymbol{y}; \boldsymbol{W}) = \frac{d_x}{2} \sum_{l=1}^{L} \alpha_l \left( \frac{1}{d_x}\boldsymbol{x}^\top \boldsymbol{a}_l \right)^2 + \frac{d_y}{2} \sum_{m=1}^{M} \beta_m \left( \frac{1}{d_y}\boldsymbol{y}^\top \boldsymbol{b}_m \right)^2 + \sqrt{d_x d_y} \sum_{n=1}^{N} \gamma_n \left( \frac{1}{d_x}\boldsymbol{x}^\top \boldsymbol{c}_n \right) \left( \frac{1}{d_y}\boldsymbol{c}_n^\top \boldsymbol{y} \right) \\
+ b_x d_x \left( \frac{1}{d_x}\boldsymbol{x}^\top \boldsymbol{1}_{d_x} \right) + b_y d_y \left( \frac{1}{d_y}\boldsymbol{y}^\top \boldsymbol{1}_{d_y} \right).
\end{aligned}
$$

This expansion constitutes a direct extension of the calculations in Appendix A, with the terms $\boldsymbol{x}^\top \boldsymbol{1}_{d_x}/d_x$ and $\boldsymbol{y}^\top \boldsymbol{1}_{d_y}/d_y$ augmented by overlaps with the eigenvectors, $\{\boldsymbol{x}^\top \boldsymbol{a}_l/d_x\}_l$, $\{\boldsymbol{y}^\top \boldsymbol{b}_m/d_y\}_{m=1}^{M}$, and $\{\boldsymbol{x}^\top \boldsymbol{c}_n/d_x, \boldsymbol{y}^\top \boldsymbol{c}_n/d_y\}_{n=1}^{N}$. By performing analogous calculations, we find that the free energy density, without assuming analytic continuation, aligns with the analytically continued free energy density. Furthermore, this free energy density is characterized by the saddle point condition involving $L + M + N = \mathcal{O}_{d_x, d_y}(1)$ variables, as in Eq. (23). If we assume $L, M, N = \mathcal{O}_{d_x, d_y}(d_x)$, then as $d_x \to \infty$, the limit becomes trivial; Without appropriate scaling, the free energy density diverges.

## C  Evaluation of Functions of the Optimal Value of Min-Max Problems

In this section, we present a method for evaluating the expected value $\mathbb{E}_A[G(\bar{\boldsymbol{x}}(A))]$ over a set of problem instances $A$, where $\bar{\boldsymbol{x}}(A) = \operatorname{argmin}_{\boldsymbol{x}\in\mathcal{X}}\max_{\boldsymbol{y}\in\mathcal{Y}} V(\boldsymbol{x}, \boldsymbol{y}; A)$ represents the min-max optimal solution. This analysis provides insights into how Eq. (2) functions as a generating function. We also demonstrate that this approach can be directly applied to evaluate the generalization error in Section 5, particularly Eq. (13).

The key idea is to expand $\mathbb{E}_A[G(\bar{\boldsymbol{x}}(A))]$ as follows:

$$
\begin{aligned}
\mathbb{E}_A\left[G(\bar{\boldsymbol{x}}(A))\right] &= \mathbb{E}_A\left[\lim_{\beta_{\min}\to+\infty}\lim_{\beta_{\max}\to+\infty}\int p_{\beta_{\min},\beta_{\max}}(\boldsymbol{x}; A)G(\boldsymbol{x})d\boldsymbol{x}\right] \\
&= d_x\mathbb{E}_A\left[\lim_{\beta_{\min}\to+\infty}\lim_{\beta_{\max}\to+\infty}\frac{\partial}{\partial\omega}f(\beta_{\min}, \beta_{\max}; \omega G(\boldsymbol{x}))\Big|_{\omega=0}\right],
\end{aligned}
$$

where we extend the free energy from Eq. (2) by introducing a parameter $\omega$ as follows:

$$
f(\beta_{\min}, \beta_{\max}; \omega G(\boldsymbol{x})) = -\frac{1}{\beta_{\min}d_x}\mathbb{E}_A\log\int d\boldsymbol{x}\exp\left(-\frac{\beta_{\min}}{\beta_{\max}}\log\int d\boldsymbol{y}\exp\left(\beta_{\max}V(\boldsymbol{x}, \boldsymbol{y}; A)\right) + \omega G(\boldsymbol{x})\right).
$$

We employ this technique to derive the generalization error in Section 5. Specifically, the generalization error for GANs can be expressed as:

$$
\varepsilon_g(\bar{\boldsymbol{w}}, \boldsymbol{w}^*) = \frac{1}{d}\mathbb{E}_\mathcal{D}\left[\|\bar{\boldsymbol{w}}(\mathcal{D}) - \boldsymbol{w}^*\|^2\right].
$$

To compute this, we augment the free energy calculation by adding the term $\omega(\|\boldsymbol{w}\|^2 - 2\boldsymbol{w}^\top\boldsymbol{w}^*)$. This adjustment is incorporated into the calculation by modifying the term $\tilde{\lambda}(w^a)^2$ in the exponent of the $d$-th power expression in Eq. (14) to $(\omega + \tilde{\lambda})(w^a)^2 - 2\omega(w^*w^a)$. Since these terms are quadratic, the Gaussian integration remains straightforward. The remaining calculation follows the same procedure in the main text and is thus omitted for brevity. Eventually, we obtain the following form:

$$
\lim_{d\to+\infty}\varepsilon_g = 1 - 2A(\hat{q}, \hat{\chi}, \hat{m}, \hat{b}) + A^2(\hat{q}, \hat{\chi}, \hat{m}, \hat{b}),
$$

where $A(\hat{q}, \hat{\chi}, \hat{m}, \hat{b})$ is determined by Eq. (22), explicitly given as

$$
A(\hat{q}, \hat{\chi}, \hat{m}, \hat{b}) = \left(\frac{\hat{b} + \sqrt{\tilde{\lambda}(\hat{q} + \lambda)}}{\hat{b}^2 + \tilde{\lambda}(\hat{q} + \lambda)}\right)\sqrt{\hat{m}^2 + \hat{\chi}}.
$$

