# OpenReview forum: "Statistical Mechanics of Min-Max Problems"
_TMLR — Accepted by TMLR_

### Review · Reviewer_rYy9 · 2024-10-09

**Summary Of Contributions:**

The paper introduces a novel statistical mechanics formalism for analyzing equilibrium values in high-dimensional min-max optimization problems, particularly addressing non-convex scenarios where the order of operations between min and max cannot be interchanged. This framework is applied to two specific problems. First, the authors provide a sensitivity analysis on bilinear min-max games to validate the model. Second, they analyze the relationship between training data and generalization error in GANs, and deriving the optimal ratio of fake to real data for training efficiency.

**Audience:**

Yes

**Claims And Evidence:**

Yes

**Requested Changes:**

Adding experimental validation, particularly in the context of GANs, would be important to make a convincing case for the applicability of the framework. One example could be to test the predicted optimal fake-to-real data ratio in a GAN for a real world task, and to show how the theoretical results translate to real-world outcomes.

**Strengths And Weaknesses:**

Strengths
- The paper introduces a novel way of looking at min-max problems by framing them within a statistical mechanics formalism. The use of a two-temperature system to model the interaction between min-max operations is interesting. The paper’s formalism opens the door for broader applications beyond bilinear games and simple GANs. he paper’s formalism opens the door for broader applications beyond bilinear games and simple GANs.
- The mathematical analysis is rigorous and well presented.
- The analysis of GAN generalization performance is a particularly strong aspect of the paper. The authors provide a clear theoretical basis for understanding how the amount of training data and the ratio of fake to real data impact GAN learning.

Weaknesses:
- The dense mathematical exposition, combined with the lack of intuitive explanations or visualizations might make the results less accessible to readers from applied fields.
- While the mathematical findings are solid, the paper would greatly benefit from empirical experiments—especially in the context of GANs. Testing the predicted optimal fake-to-real data ratio and evaluating the framework’s performance on real-world problems would significantly strengthen its impact.
- The GAN model used is quite simplified, and applying the formalism to more realistic GAN architectures would make the results more relevant to current research trends. This would be helpful to showcase the full potential of the framework in real-world machine learning applications.

---

> ### Author Response · Authors · 2024-11-04
> **Response to Reviewer rYy9**
>
> We appreciate your recognition of the novel formulation we introduced for analyzing min-max problems and its potential for future applications.
> We thank you for your positive feedback on the mathematical rigor of our approach.
> In response to your insightful review, we have made several improvements, outlined below.
>
> **Enhanced Explanations and Step-by-Step Calculations**
>
> To make our work more accessible to practitioners, we have expanded the intuitive explanations and provided more detailed, step-by-step derivations for key results.
>
> **GAN Application and Theoretical Focus**
>
> We agree that empirical validation of GANs is valuable. However, our main contribution lies in presenting a new theoretical approach to the sensitivity analysis approach of general min-max problems.
> In this paper, we applied our formulation to conduct a theoretical analysis of both bilinear min-max games and a simplified GAN model as examples.
> Although simplified compared to real-world GANs, these models still present notable theoretical challenges.
> Therefore, we hope the theoretical advances presented here will be recognized as our primary contribution, even without extensive experimental validation.
>
> In response to your comments, however, we conducted additional experiments using 4-layer perceptrons with various batch sizes, 64 and 128, on MNIST generation tasks to explore the dependency of FID on varying ratios, $r$.
> While we observed fluctuation in outcomes based on random seeds and batch sizes, no significant trends were established.
> These results suggest the need for further research to understand the dynamics of a min-max learning algorithm.
> We also acknowledge that our model class may be overly simplistic and that expanding our work to encompass more complex architectures presents an intriguing direction for future research.
> If desired, we are prepared to share results from experiments with the fixed batch sizes that analyze FID as a function of the ratio, $r$, in the MNIST task; however, it remains unclear whether these preliminary numerical experiments should be included in the main text.

---

### Review · Reviewer_W6Mb · 2024-10-13

**Summary Of Contributions:**

This paper introduces a statistical mechanical formalism for analyzing equilibrium points of min-max problems in a high-dimensional perspective. Moreover, the authors apply the above formalism to the study of bilinear min-max problems and GANs, for which numerical experiments have also been conducted.

**Audience:**

Yes

**Broader Impact Concerns:**

I do not have any concerns on the ethical implications of this paper.

**Claims And Evidence:**

Yes

**Requested Changes:**

In this paper, I can emphasize the following shortcomings:

> **Intepretabilty**

$\bullet$ I would suggest to relocate section **Related work** from the end to the beginning of the paper; to be more precise, after **Introduction**. The main reason for this is that the **Related work** section allows you to understand what existing results the authors are relying on in their paper, and in my opinion, it should be before the main results.

$\bullet$ The authors mentioned that the presented formalism is investigated under high-dimensional consideration. It may be not quite clear to the reader why the formalism is given exactly for high-dimensional problems. It may be worth adding a comment on this matter in the text of the paper.

$\bullet$ As for me, it is not clear how replica trick allows to move from Eq. (5) to Eq. (6). As I know, the replica trick consists in the following expression:
$$\ln Z = \lim_{n \to 0}\frac{Z^n - 1}{n}.$$
Could you explain this transition, please?

> **Сleanliness of text writing**

$\bullet$ Comma instead of dot on page 1:  "$\ldots$loss function (Szegedy et al., 2013; Goodfellow et al., 2014b; Papernot et al., 2016; Madry et al., 2017)$\textcolor{red}{,}$"

$\bullet$ Missing comma in formula on page 2.

$\bullet$ Superfluous numbering of formulas: I have not found any references on Eq. (12) and Eq. (24). There may be other formulas with redundant enumeration.

$\bullet$ There are different designations for definitions. For example, sometimes the authors use $\stackrel{\triangle}{=}$, and sometimes just $=$, for example in Eq. (12). It would be better to unify the style of definitions.

$\bullet$ Extra dot on page 4: "The detailed proof of this theorem is provided in Appendix$\textcolor{red}{.}$ A."

$\bullet$ Missing comma on page 4 in Theorem 3.1.

$\bullet$ Overflow of formulas on pages 14-16.

There may be other shortcomings, so I'd suggest to authors to double-check the entire paper.

It is worth noting that the **Interpretability** section is more important in terms of understanding the work, while the **Cleanliness of text writing** is a review for minor blunders.

**Strengths And Weaknesses:**

**Strengths**

> Novelty

The technique presented is indeed new compared to the existing ones. Also, this work inspires a more detailed study of the equilibrium properties of min-max problems.

> Mathematical calculations

I found no errors in the deductions made by the authors.

**Weaknesses**

> Intepretability

There are some parts of the work that I find difficult to interpret. This makes it noticeably more difficult to understand the contribution of the paper. For more details see **Requested Changes**.

> Сleanliness of text writing

I noticed some marks that affect the cleanliness of the text of the paper. For more details see **Requested Changes**.

---

> ### Author Response · Authors · 2024-11-04
> **Response to Reviewer W6Mb**
>
> We sincerely appreciate your recognition of the novelty of our min-max problem formulation and its applications to GANs, as well as your encouragement to explore sensitivity analysis of equilibrium solutions in min-max problems as a promising area for future research.
> Additionally, we are grateful for your insightful review, which has guided our revisions.
>
> **Enhanced Placement of Related Work**
>
> In response to your suggestion, we have moved the Related Work section immediately after the Introduction. This rearrangement clarifies the existing context before presenting our contributions and main results.
>
> **Clarification of Replica Method**
>
> In response to your request for clarification on the derivation from Eq. (2) to Eqs. (4) and (5), we have provided a more detailed explanation in the revised manuscript.
> Specifically, we demonstrate that Eqs. (4) and (5) can be derived by applying the identity in Eq. (3), as shown in the revised manuscript.
> Furthermore, Eq. (3) can be derived by applying a Taylor expansion.
> We believe these additions will help interpret the mathematical steps, particularly for readers less familiar with the replica method.
>
> **Explanation of High-Dimensional Limits**
>
> Our results hold in the high-dimensional limit because the saddle point method becomes exact in this limit.
>
> **Text Cleanliness**
>
> We greatly appreciate your detailed feedback on text clarity and typographical issues.
> We have carefully addressed each specific point you raised, including missing or misplaced punctuation, redundant formula numbering, and inconsistent variable notations.

---

### Review · Reviewer_oYGi · 2024-10-22

**Summary Of Contributions:**

The authors propose using a heuristic from statistical mechanics, namely the replica trick, to study min-max problems. They use the Boltzmann distribution in the limit to describe the min-max-value and the replica trick in essence allows for exchanging expectation and logarithm to simplify computation.

**Audience:**

Yes

**Broader Impact Concerns:**

I do not have any concerns on the ethical implications of the work that would require adding a Broader Impact Statement.

**Claims And Evidence:**

No

**Requested Changes:**

Typos
------
- In the notation section it should be d a^i instead of d a_i, because that is how it is used in the sequel if I am not mistaken.
- In the sentence after eq. (1) it should say "e.g. *a* graph $G$".
- At the beginning of the first line of p. 3 there is an "is" missing.
- In eq. (6) in the exponent the "; A" should be included in the round brackets.
- At the beginning of p. 4 it says "$V(\cdot, \cdot)$ is given by $V(x, y; W)$ ..."
- Section 4.1: it should say "drawn *from* the following distribution" and "*are" the latent values of the fake data".
- Section 4.2: the expression Z_gamma is missing the $(\beta_\min, \beta_\max)$ (at least that is how it is written in other parts of the paper).
- Line before eq. (23): it should say "*we* obtain".
- Figure 1. "*Generalization* error"
- In the first formula in appendix A the expectation should be over W instead of over A.
- "non-convex to the variables" sounds off, I instead suggest "with respect to $x, y$" or something similar.
- In (7) the dependence of $V$ on $A$ is missing.
- In the last line of the big equality chain on the first page of the appendix, should it be $+$ instead of $\times$ separating both terms?

Bibliography
-------
- Leaving out the third author, Jean Barbier, in the second reference and then using "et al" seems strange. This reference is also missing pages numbers.
- Names of conferences and journals, like ICML, NeuRIPS, Physical Review Letters, or Princeton University Press should be capitalized.
- In the fifth reference, the first names of the authors are abbreviated by their first letter, which is inconsistent with the other references. Something similar is true for the paper by Nemirovsky et al. and by Papernot et al. Similarly, Ian Goodfellow is once listed like that and once as Ian J Goodfellow.
- In the seventh reference, the name of the journal is abbreviated, which is inconsistent with the other references.
- Mathematician's names or common abbreviations like GNNs, PCA, VAE, Boltzmann or Hilbert should be capitalized, like they are in the titles of these articles.
- The reference "Learning dynamics in linear VAE: ..." is the only one with a URL attached, which is inconsistent.

Formatting
-------
- I suggest replacing $e^{...}$ by $\exp\left(...\right)$ in almost all instances to enhance readability, because the ... will be too small otherwise.
- I also think it would benefit to have brackets indicate where the arguments of e.g. the natural logarithm or an expectation end, and it is clear if exponents are outside or inside of these operators.
- The notation $p(c) = \mathcal{N}(0, 1)$ and its cousins used in section 4.1. are not clear to me, since the right hand side does not depend on $c$. If I see this correctly, $p(c)$ should be the value of the density of a N(0, 1)-distributed random variable at the point $c \in \mathbb{R}$.
- Omitting indices and regions of integration declutters the notation but makes it harder to see what is going on, especially when reading through this submission for the first time. For example, in the first big equation in subsection 4.2, is the integration over $\mathcal{X} \times \mathcal{Y} = \{ 0, 1 \}^{d_x + d_y}$, as suggested by equation (7), or over $\mathbb{R}^d \times \mathbb{R}^d$?
- Using $d$ for both the discriminator, which takes input variables, and the dimension (see notation section) can cause confusion.
- I am assuming that equation (4) is a definition, which is usually marked with $\overset{\triangle}{=}$ in the submission, but since this is not carried out consistently - as far as I can tell - I am not sure, but this matters. I have the same issue with equation (7).
- I do not understand why equation (2) is phrased more complicated than necessary, why can the simplification that is preformed in eq. (5) be applied to cancel the exponential and the logarithm?

**Strengths And Weaknesses:**

Main review
--------
- I agree that, in principle, the papers main idea can be useful. However, at least to me, the mathematical derivations are not clear, which is why I recommend only accepting this paper if substantial revision have been made. To be more precise: pretending that parameters $p$ and $\gamma$ are integers and that the entries of the vectors in question are iid (that is, we replicate a 1d-particle d times) one can pull the exponent out of the expectation and together with the limit formula for the natural logarithm exchange expectation and logarithm. But, then taking $\gamma \to 0$ conflicts with the assumption that $\gamma$ is a integer. However, no bounds / estimation errors or any of the like are presented, so for an non-expert like me it is very difficult to gauge if this heuristic is meaningful. The only concrete evidence for the efficacy of this idea is provided in the proof of Thm. 3.1, however it seems to me that the calculations performed there also hinge on the fact that the parameters are integers, so that, in effect, nothing is proven.
- Furthermore, if I understand correctly, mathematical equality and heuristic are mixed, e.g. (6) should not be an equality, but the right hand side is just an heuristic to the left side, right?
- Equations (3) and (1) are contradictory since $\Omega$ depends on $A$, however, the right hand side of (3) does not.
- It would be nice if there were so hints given, e.g. in the appendix, on how the claimed extension of "scalar" $W_{xx}$ to matrices with limited number of eigenvalues (what does "of $\mathcal{O}(1)$ mean here?) can be accomplished.
- Furthermore, I do not understand, why in going from (14) to (15), there is no $f$ in the latter formula. Naively plugging in (12) into (14) gives a different result than (15).
- Am I assuming correctly that in subsection 4.2 we always consider $\gamma, p \in \mathbb{N}$ and not use implicitly somehow "analytic continuation". If yes, this should be clearly stated at the beginning of the subsection.
- What are the $Dc$ and $Dz$ in the equation defining $\mathcal{T}$ and is the integration over $\mathbb{R}^{p \gamma}$?
- I think that the self-consistent (what does this mean, exactly?) equations after eq. (27) are not quite correct: the signs of b and \hat{b} should be reversed and we should instead have $\hat{\chi} = - \frac{\alpha \eta (q \eta + m^2)}{(\eta \chi - 1)^2} - \frac{\tilde{\alpha} \tilde{\eta} (q \tilde{\eta} + b^2)}{(\tilde{\eta} \chi + 1)^2}$.
- To be honest, I do not know how $\varepsilon_g$ was derived and would appreciate a bit more detail on that.
- For people not too familiar with statistical mechanics I recommend providing more background, maybe in form of a short and concise subsection (clearly defining the terms "$\extr$, "(virtual) two-temperature system", "randomized instance ensembles", "dataset ensembles", "complete rationality", "handling the integral using the saddle point method") and adding more references, e.g. for backing up the statement that "Such an approach generally does not yield accurate results when the function $V$ is non-convex" or the imprecise claim that "the distribution $\lim_{\beta_{\max} \to \infty} \lim_{\beta_\min} ...$ concentrates on a uniform distribution over the min-max values" (this statement is not precise regarding the type of convergence and probably needs some assumptions on $V$).

---

> ### Author Response · Authors · 2024-11-04
> **Response to Reviewer oYGi**
>
> Thank you for your insightful review and for recognizing the utility of our min-max formulation. We have addressed all points raised in your review and revised our manuscript accordingly. Below, we provide a point-by-point response summarizing the modifications made to clarify and strengthen our work, following your recommendations. Please refer to the updated manuscript for detailed modifications.
>
> **Mathematical vs. Heuristic Equations**
>
> We apologize for any confusion regarding the distinction between mathematical and heuristic equations in our analysis.
> In this revision, we have clearly distinguished cases where equations are mathematically exact.
> While there may have been a misunderstanding, you may have viewed Eq. (5) in the revised version as not exact.
> However, Eq. (5) in the revised version can be rigorously derived using the identity in Eq. (3).
> At this stage, we treat $p$ and $\gamma$ as real numbers.
> In the replica analysis, we initially assume $p$ and $\gamma$ are natural numbers, as shown in Eq. (6), and then take the limit as $p$ and $\gamma$ approach 0 by analytic continuation to obtain the result. At this stage, mathematical rigor is not guaranteed. However, as noted in related work, this analytic continuation has yielded valuable results in information science, including machine learning, without significant issues.
>
> Additionally, in the revised Section 4, we demonstrate that evaluating the exact free energy in Eq. (5) with $p$ and $\gamma$ as real numbers yields results consistent with those obtained by evaluating Eq. (6) with $p$ and $\gamma$ as natural numbers, specifically in the case of bilinear min-max games, a fundamental example of a min-max problem. Therefore, we have demonstrated the validity of this heuristic method, at least in this fundamental case.
>
> **Consistent Mathematical Representation**
>
> To avoid further confusion, we have ensured that only mathematically rigorous expressions are presented as equalities throughout the revised manuscript.
>
> **Extension of Theorem 3 and Notation for $O_{d_{x}, d_{y}}(1)$**
>
> We have extended the discussion of Theorem 3 to include cases where the matrices $W_{xx}$, $W_{xy}$, and $W_{yy}$ have $O_{d_{x}, d_{y}}(1)$ eigenmodes in Appendix B. Additionally, we clarified the definition of $O_{d_{x}, d_{y}}(1)$ in the notation section.
>
> **Clarifications on Section 5**
>
> Following your suggestion, we have added explanations of the transformations related to $V(w, v; D, \tilde{D})$ in Eq. (12) for GANs to improve clarity.
> Additionally, in the analysis of GANs in Section 5, we note that providing a rigorous justification for analytic continuation is challenging, unlike for the bilinear min-max games in Section 4.
>
> **Additional Notational Clarifications**
>
> We have clarified definitions for $Dz$ and $Dc$ in the notation section.
>
> **Verification of Self-Consistent Equations**
>
> Following a thorough review, we confirmed that the self-consistent equations derived from the gradient of the free energy are correct.
> These equations are derived from calculating the gradient of the free energy density in the manuscript.
>
> **Evaluation Method for General Optimal Value Functions**
>
> To clarify, we have added specific details of the method used to evaluate general functions of the optimal values, including the generalization error $\varepsilon_{g}$ in Appendix C.
>
> **Detailed Explanation of Specialized Terms**
>
> We have provided more detailed explanations for terms such as $\mathrm{extr}$ and added definitions for other specialized terms to enhance accessibility for readers.
>
> **Addressing Minor Errors and References**
>
> Thank you for highlighting the typos and reference inconsistencies. These issues have been addressed in the revised manuscript, with citations and formatting corrected accordingly.

---

### Decision · Action_Editor_c47M · 2024-12-04

**Recommendation:** Accept as is

**Comment:**

All the reviewers acknowledged that the paper provides a novel technique for analyzing min-max problems using statistical mechanical formalism. There is also a consensus among the reviewers that the paper should be accepted. The authors also improved the readability during the discussion and

**Audience:**

The problem considered in the paper is relevant to TMLR's audience. I believe many individuals will be interested in the findings of this paper.

**Claims And Evidence:**

The claims made in the submission are supported by accurate and clear evidence, as all reviewers acknowledged. In the revision, the author also adequately addressed the reviewers' requests, and all reviewers recommended the acceptance.